# Preliminary Study: Depriving Piglets of Maternal Feces for the First Seven Days Post-Partum Changes Piglet Physiology and Performance before and after Weaning

**DOI:** 10.3390/ani9050268

**Published:** 2019-05-23

**Authors:** Edgar O. Aviles-Rosa, Anoosh Rakhshandeh, John J. McGlone

**Affiliations:** 1Laboratory of Animal Behavior, Physiology and Welfare, Animal and Food Sciences Department, Texas Tech University, Lubbock, TX 79409-2141, USA; edgar.aviles-rosa@ttu.edu; 2Department of Animal and Food Sciences, Texas Tech University, Lubbock, TX 79409-2141, USA; Anoosh.rakhshandeh@ttu.edu

**Keywords:** coprophagy, behavior, pig performance

## Abstract

**Simple Summary:**

Coprophagy is the behavior of eating feces. This behavior has been reported in many animals, including pigs. We aimed to assess how coprophagy affects piglet behavior, physiology, and performance by depriving piglets of maternal feces for the first 7 d post-partum. Eight litters were randomly assigned either to have access to maternal feces (control) or to be deprived of maternal feces for the first 7 d post-partum. Piglet behavior was observed for 24 h at 7 d of age and two piglets from each litter (male and female) were bled for hematological analysis at 0, 7, and 21 d of age. Piglets feed intake and weight gain were measured until 123 d post-weaning. No behavioral differences were observed between treatments. Overall, control piglets had 25% higher white blood cell counts and higher feed intake and weight gain than piglets deprived of maternal feces. At 123 d post-weaning, control piglets were 9.6 kg heavier than piglets deprived of maternal feces. In conclusion, piglets with access to maternal feces early in life exhibited better performance. The mechanisms by which coprophagy improves piglet performance needs to be elucidated in further studies, but could include effects of the nutrition, microbiome or semiochemical exposure.

**Abstract:**

Coprophagy has been described in piglets although its importance has not been fully assessed. The aim of this study was to evaluate how deprivation of maternal feces influenced piglet physiology, behavior, and performance. Eight litters were randomly assigned to one of two treatments. Control (CON) litters had access to maternal feces while deprived (DEP) litters were deprived of maternal feces for the first 7 d post-partum. Piglet behavior was quantified for 24 h at 7 d of age. Blood samples were collected from one male and female from each litter at 0, 7, and 21 d for hematological analyses, and post-weaning performance was assessed until 123 d post-weaning. No treatment effects were observed on piglet behavior. DEP piglets had 25% lower leukocyte counts (*p* < 0.01). Relative to DEP litters, CON litters had increased post-weaning feed intake (0.998 vs 0.901 kg/d; *p* = 0.02) and weight gain (0.536 vs 0.483 kg/d; *p* < 0.01). At 123 d post-weaning, CON pigs were 9.3 ± 2.3 kg heavier than treatment pigs (*p* < 0.01). These results suggest that access to maternal feces improves immunocompetence and growth performance. Further studies are needed to explore the physiological mechanisms through which maternal feces improve growth performance, including nutritional and microbial factors, or the presence of maternal semiochemicals.

## 1. Introduction

Coprophagy is the behavior of eating feces [1]. This behavior has been reported in insects, rodents, lagomorphs, and other mammals including horses, swine, and some non-human primates [1,2,3]. Coprophagy may provide an additional source of bioavailable energy and other nutrients and minerals such as protein, volatiles and non-volatile fatty acids, Zn, Cu, and Fe, especially to newborn animals [1,2,3]. Coprophagy is also a natural mechanism by which animals establish the intestinal microbiota essential to intestinal development and the process of digestion [1,2,3,4]. For instance, foals obtain vitamins, enzymes, and nutrients to compensate for nutritional deficiencies and essential bacteria for cellulose digestion via coprophagy [2,4]. Similarly, coprophagy benefits rat pups since it results in the consumption of bile acids that aid the myelination process and help guard the gut against pathogenic bacteria. [5]. Furthermore, rat pup mortality due to enteritis increased when access to maternal feces was restricted [6]. It has been estimated that nursing piglets on average eat 20 g/d of maternal feces [7]. Different studies speculated that piglets (especially those kept outdoors) avoid anemia by obtaining iron from dirt and by eating maternal feces [7,8,9,10,11,12]. Nonetheless, these studies lack an experimental group deprived of maternal feces. Furthermore, the effects of coprophagy on piglet behavior, hematology, and growth performance have not been evaluated to date. Therefore, the aim of this preliminary study was to evaluate the effect of coprophagy on these parameters by depriving piglets of maternal feces. Based on the information discussed above, we hypothesized that piglets deprived of maternal feces would become anemic and will have a lower growth rate compared to the ones that had access to maternal feces. 

## 2. Materials and Methods 

### 2.1. General

This study was conducted at the Texas Tech University (TTU) Swine Research Farm as a part of our on-going work on swine semiochemicals. The TTU Animal Care and Use Committee (ACUC) reviewed and approved all procedures in this study (protocol # 16105-11). Eight litters from PIC Camborough (Pig Improvement Company, Hendersonville, TN) sows (third parity) were randomly assigned to one of two treatments (*n* = 4) following a completely randomized design (CRD). Litters were housed in conventional slatted floor farrowing crates (1.52 m × 2.13 m). Sows were fed 6.8 kg/d of a lactation corn-soybean meal (SBM) based diet (Table 1.) from two weeks before the expected farrowing day until weaning. No creep feed was given to piglets during the experiment. 

### 2.2. Treatments and Nutritional Analysis

Control (CON) litters (*n* = 4) had free access to maternal feces, and experimental group litters (deprived group; *n* = 4) were deprived (DEP) of maternal feces from birth until 7 d of age. Fecal deprivation was until 7 d of age since piglets will show signs of anemia at this point in time if no extraneous iron had been provided. Ten days before the expected farrowing day, sows were moved to individual farrowing crates. Until farrowing, all crates were washed daily with water to remove maternal feces. From farrowing until 7 d of age, an observer was on the farm 24 h/d weighing maternal feces immediately after defecation to determine sow fecal output and removing maternal feces from the pens of DEP litters. After being weighed by the observer, maternal feces of CON litters were placed over a rubber pad in a corner of the pen to prevent sows from blocking piglet access to feces, as well as to prevent feces from falling into the pit (Figure 1). Fecal samples (5 g) from all sows were collected once a day (at 6 a.m.) and pooled for the duration of the study, oven dried at 55 °C for 48 h, and used for determination of the nutrient contents. Dietary and fecal sample nutrient analyses were performed at the Dairy One, Inc. laboratory (Ithaca, NY). Fecal true protein (TP) concentration was measured using the bicinchoninic acid (BCA) assay kit (Thermo Fisher Scientific, Bellefonte, PA) according to the manufacturer’s instructions. After 7 d of age, piglets in both treatment groups had free access to maternal feces. Piglets were ear notched, teeth clipped, injected with iron dextran and penicillin, and then castrated (males only) at 7 d of age.

### 2.3. Pre-Weaning Performance, Blood Collection and Behavior

Pre-weaning survival and mortality rates, body weight (BW), and average daily gain (ADG) of piglets were evaluated at 0, 7, and 21 d of age. Blood samples (~1 mL) were collected into K_2_ EDTA vacutainer tubes from one male and female piglet from each litter via jugular venipuncture at 0, 7, and 21 d of age. Piglets were selected at random from those with body weights close to the average body weight of the litter. An automated cell counter (Vet Scan HM 5; Abaxis) was used to evaluate hemoglobin (Hb), hematocrit (HCT), mean corpuscular volume (MCV), red blood cells (RBC), and white blood cells (WBC). The neutrophil to lymphocytes (NLR) ratio was calculated for the data obtained. Hematological values from both piglets within a litter were averaged. The average value was considered to be a representative value of the litter and used for the statistical analysis.

Piglet behaviors including nursing, laying, fighting, and active behaviors were evaluated for 24 h on the 7th day of age, the last day of fecal deprivation. In addition, the time at least one piglet from the litter was observed interacting with maternal feces was recorded for control litters. For this, video cameras were placed over the farrowing crates to record the behavior of the entire litter. Definitions of behavioral observations are given in Table 2. To evaluate high frequency behaviors (laying, active, standing) a trained observer conducted a ten-minute scan sample for 24 h. A continuous sampling technique was used to quantify low frequency behaviors such as interaction with feces, fighting and nursing behaviors. For each litter, the duration of a nursing bout, bout interval, and the number of bouts per hour were recorded. All behavioral data were observed by a single observer and no observation software was used to code behavioral data. 

### 2.4. Post-Weaning Performance

Piglets were weaned at 25 ± 2 d of age. To keep the experimental unit intact, at the time of weaning, the piglets from each litter were kept together and moved to nursery pens. Raised pens measuring 1.5 × 3 m, with slat plastic floor were used as nursery pens. Piglets in both treatments were housed in the same barn. After weaning, average daily feed intake (ADFI), ADG, and feed-to-gain ratio (F:G) were measured weekly for a period of four weeks. ADFI was calculated by estimating weekly feed consumption by weighing the remaining feed on the feeder. ADG was calculated by weekly changes in piglets body weight. F:G was the ratio between ADFI and ADG. Four weeks after weaning, litters were moved to the farm growing-finishing facility as described above. Finishing pens were each 3.65 × 2.15 m with a slat concrete floor. Measures of growth performance (ADG, ADFI, and F:G) were evaluated monthly until 123 d post weaning. During the post-weaning period, pigs were fed corn-SBM based diets according to the phase-feeding program.

### 2.5. Statistical Analysis

Data were analyzed as a CRD in SAS (SAS version 9.4; SAS Inst., Inc., Cary, NC, USA). Litter was considered the experimental unit (*N* = 8). Normality and homogeneity of variances were confirmed using the univariate procedure (PROC UNIVARIATE). A repeated measure analysis of variance (PROC GLIMMIX) was used to evaluate parameters measured over time, such as blood cell counts and performance data. The model included the effect of time, treatment, and their interaction as fixed effects and litter within treatment (experimental unit) as a random effect. Litter size (LS) was used as co-variate. A first order auto regressive co-variance structure was included in the model since it produced the smallest AIC value. Fisher’s least significant difference (LSD) was used to determine any treatment effect within the period. A one-way ANOVA was used to evaluate litter behavior at 7 d of age. The effect of days post-partum on sow fecal output was evaluated using the Friedman test since data did not meet parametric assumptions. To simplify our discussion, the presented means and standard errors (SE) correspond to the raw data and not to the ranks. Differences were considered significant when *p* ≤ 0.05. 

## 3. Results

### 3.1. Sow Fecal Output and Nutreint Content

The nutrient contents of sow fecal samples are presented in Table 3. Fecal output was not affected by treatment. Sow fecal output (DM basis) was significantly affected by post-partum day (T*_7,5_*_5_ = −2.72; *p* < 0.01). None of the sows defecated on the day of farrowing and only two and three sows defecated on day one and two post-farrowing, respectively. All sows defecated daily after 3 d post-partum. Fecal output was significantly lower at farrowing and at days one and two post-partum relative to the other days (*p* < 0.01). Fecal output the third day post-partum did not differ from 5, 6, and 7, but was higher than day four fecal output (Figure 2). On average, sows defecate 216 ± 24.8 g of DM daily from day 3 to 7 post-partum.

### 3.2. Measures of Behavior 

Data from one litter in the CON group was excluded from analysis due to problems with the recording system. No treatment effects were observed on measures of litter nursing, laying, active, standing, or fighting behaviors at 7 d of age (Table 4.). Behavioral data showed that at 7 d of age, piglets within a litter spent 4.27 ± 0.51 min interacting with maternal feces per day. 

### 3.3. Piglet Hematology and Survival

Table 5 shows CON and DEP litters pre-weaning hematology and mortality and survival rates. Litter size (LS) was not affected by treatment. A co-variate effect on mortality rate, survival rate, WBC, and neutrophil (NEU) counts, as well as the neutrophil lymphocyte ratio (NLR) was observed (*p* < 0.05). There was a significant treatment effect for WBC counts (F_1,6_ = 16.05; *p* < 0.01). The WBC count was higher in CON pigs than in DEP pigs during the first 21 d of age. A time effect (*p* < 0.05), was found for all blood parameters but not for litter survival rate and WBC. Red blood cells, Hb, MCV, and HCT levels were lower at 7 d of age than at birth (*p* < 0.05), but at 21 d of age, RBC, Hb, and HCT values were higher than they were at birth. At 21 d of age, MCV values were similar to the MCV observed at birth. Lymphocytes (LYM) and monocytes (MON) blood counts increased at 7 d of age (*p* < 0.05). No difference was observed between LYM counts at 21 and 7 d of age. Monocyte counts at 21 d of age were similar to the ones observed at birth. The interaction effects between treatment and time was significant for NLR (F_2,11_ = 27.54; *p* < 0.01). At birth, control piglets had higher NLR than DEP piglets (4.69 vs 2.15 ± 0.21), but no differences were observed after birth.

### 3.4. Pre and Post-Weaning Growth Performance 

Piglet pre- and post-weaning growth performance data are presented in Table 6. The main effect of treatment was significant for piglet ADG (F_1,6_ = 20.48; *p* < 0.01) and post-weaning ADFI (F_1,6_ = 8.52; *p* = 0.02). Piglets in the control group were significantly heavier than DEP piglets after 56 d post-weaning. At 123 d post-weaning, control piglets were 9.3 kg heavier than treatment pigs (100.7 vs 91.4 ± 2.29 kg; *p* < 0.01). Overall, there was no treatment effect on the F:G ratio, but control litters had a greater ADG (0.536 vs 0.483 ± 0.008) and ADFI (0.998 vs 0.901 ± 0.023) than DEP piglets.

## 4. Discussion

The aim of the study was to evaluate the effects of coprophagy on piglet blood cell counts, behavior, and performance. Coprophagy benefits were studied by depriving piglets of maternal feces until 7 d of age. From days 3 to 7 post-partum, sows excreted close to one kilogram of wet feces or around 216 ± 24.80 g of DM per day. The low or absent fecal output observed during the first two days post-partum can be associated with the reduction in feed intake that often occurs around the time of farrowing in modern pig production units [13] since low feed intake reduces the passage rate of digesta [14]. Based on the estimated fecal output and assuming an average litter size of 10 pigs, each pig could eat 21.62 g of feces per day. This is in general agreement with Sansom and Gleed estimation of piglet fecal consumption [7]. 

We deprived piglets of maternal feces for only 7 d since at this time point piglets should be presenting signs of anemia if no external iron is provided due to their rapid growth rate, rapid increase in blood volume, and the low iron content of sow milk [15,16]. This allowed us to study the effect of fecal deprivation without affecting the piglets’ welfare. After deprivation of maternal feces, no treatment effects were found on piglet RBC, Hb, HCT, MCV, or any other blood parameters associated with the diagnosis of anemia at 0, 7, or 21 d of age. At 7 d of age, piglets in both treatment groups had RBC, MCV, HCT, and Hb, levels like the ones found in mildly anemic piglets [15,16]. In addition, behavior data showed that both groups lying behavior was higher than the ones reported in 5 d old piglets [17,18], which could be another sign of mild anemia. After day 7, erythrocyte parameters improved since all piglets received 2 mL of iron dextran. Thus, contrary to what has been reported in the past [7,8,9,10,11], coprophagy did not prevent anemia in this study. It is possible that to see any effects on RBC parameters, longer periods of deprivation are required. However, hemoglobin differences in our study did not indicate any trends, so it is unlikely that piglet fecal intake can greatly impact red blood cell or hemoglobin levels in a commercial setting.

Control litters had higher WBC counts at birth and until 21 d of age. Differences in WBC, NEU, and NLR observed at birth can be a consequence of differences in bleeding time. Due to limitations with birth times, blood samples were taken in different time points, which led to variability in piglet colostrum intake. Since three out of the four litters in the control group were bled overnight, control piglets could have higher WBC count since they had more time to consume colostrum [19]; however, we think this is unlikely. Nevertheless, control litters WBC continued to be higher until 21 d of age. This was expected since control piglets were exposed to feces for more time. By eating or being in contact with maternal feces, piglets may have internalized the maternal fecal microbiota [20,21,22,23]. The latter can lead to the development of an acquired immune system and the proliferation of WBC. These results showed that having access to maternal feces early in life might improve piglet immunocompetence.

In the current study, although having access to maternal feces did not improve piglet red blood cell parameters associated with anemia, it improved piglet growth performance. At 123 d post-weaning, control litters remained heavier and had a higher ADG and ADFI. Piglets iron intake can impact growth rate. Piglets need to consume 7 mg of iron per day [24] or 21 mg of iron per kilogram of body weight gained [16]. Because of this, heavier piglets tend to have lower values of RBC, Hb, HCT [25,26]. The greater blood volume and iron requirement of fast-growing pigs makes them prone to iron deficiency anemia [25]. Thus, deprivation of maternal feces could have limited piglet iron ingestion leading to lower growth rate in the DEP group. We are still elucidating the possible mechanisms by which having access to maternal feces early in life improved piglet growth performance. Based on our findings, we speculate that early exposure to maternal feces improved piglet growth performance through a nutritional and/or microbiological effect. In addition, the presence of maternal semiochemicals in lactating sow feces could also play a significant role in these findings. Most likely the benefits of early exposure to maternal feces on piglet growth performance is due to an interaction between these possible mechanisms. 

As fecal nutrient content analyses showed, lactating sow feces are rich in minerals (Fe, Zn, Cu, and Mn) and have a high concentration of true protein. By eating maternal feces, piglets may obtain bioavailable nutrients that promote their growth and promote gut health. For instance, feces are rich in Cu and Zn, minerals that have antibacterial and growth stimulating properties [27]. It has been shown that supplementing 250 ppm of Cu and/or Zn in weaning diets increases post-weaning performance [27,28]. Hence, we speculate that coprophagy might increase piglet mineral and nutrient intake, therefore enhancing performance and guarding the piglet gut against pathogenic bacteria. 

Another possibility is that maternal feces provided piglets with essential bacteria that are beneficial to the gastrointestinal tract, as in other species [1,2,3,4,5,6]. Since current housing conditions allow pigs to be in contact with sow skin, vulva, feces, and urine, piglet microbiome is likely to be dependent on the sow [20,21]. Microbial transfer from sow to piglets starts when piglets exit the birth canal and continues via colostrum, milk, and feces consumption [20,21]. Furthermore, during piglets first week, piglet and sow fecal metabolic fingerprints are alike [22]. Thus, fecal deprivation during the first week of age could have prevented early colonization of beneficial bacteria from sow feces. Studies in humans suggest that disruption in the early colonization of the gastrointestinal tract can have long-term health consequences [20]. In piglets, the development of the mucosal immune system is dependent on piglet microbial exposure [20,23]. Thus, we hypothesized that coprophagy may allow piglets to obtain beneficial microbes from maternal feces. These microbes may help piglets maintain a proper intestinal environment and develop the mucosal immune system thereby enhancing their performance. This might also explain why CON piglets had higher WBC counts. Future studies of healthy lactating sow fecal microbiome may result in the discovery of beneficial bacterial populations that could be used as a species-specific probiotic. 

Lastly, deprivation of maternal feces could reduce piglet growth performance by removing maternal semiochemicals from the environment. Fecal maternal semiochemicals have been reported in other species that also exbibit coprophagy behavior. For instance, rat dams secrete a maternal fecal pheromone (deoxycholic acid) which attracts pups and promotes coprophagy [5]. Similar to rats, previous studies have found that piglets have a preference for maternal feces compared to other maternal odors as early as 12 h of life. [29]. As in rat pups, this preference may be caused by maternal semiochemicals present in lactating sow feces. Maternal semiochemicals have been shown to improve weaning piglet performance. For instance, the pig appeasing pheromone and the rabbit maternal pheromone (2-methyl-2-butenal) reduced aggression and improved performance when sprayed on the feeders of weaned piglets [30,31]. Thus, it is possible that sow feces contain natural maternal semiochemicals that promote health and performance. Future studies are needed to identify maternal semiochemicals in lactating sows that could enhance piglet growth performance. 

## 5. Conclusions

In this preliminary study, the effects of fecal deprivation on piglet hematology, behavior, and growth were evaluated by depriving piglets of maternal feces for the first 7 d of life. Depriving piglets of maternal feces had no significant effect on piglet behavior. Deprivation of maternal feces reduced the piglets’ WBC counts. By having access to maternal feces early in life, piglets improved their pre- and post-weaning performance. Data suggest that early exposure to maternal feces improves piglet performance and immunocompetence. Further studies are needed to determine the mechanisms by which coprophagy improves performance and immunocompetence. We speculate that the presence of fecal maternal semiochemicals might induce coprophagy, and that by consuming maternal feces piglets may obtain nutrients and beneficial microbes that have long-term effects on their performance. 

## Figures and Tables

**Figure 1 animals-09-00268-f001:**
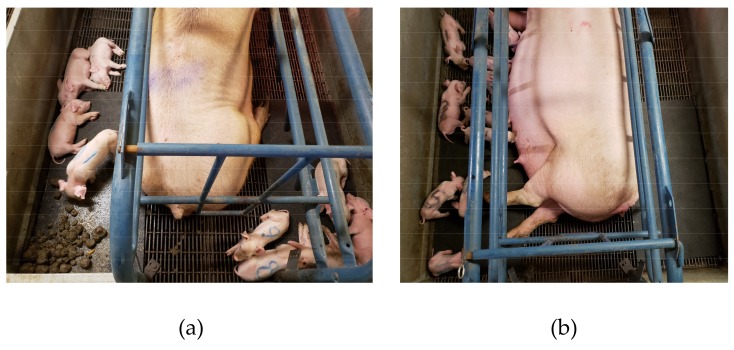
Control (**a**) and deprived (**b**) litters with and without maternal feces respectively.

**Figure 2 animals-09-00268-f002:**
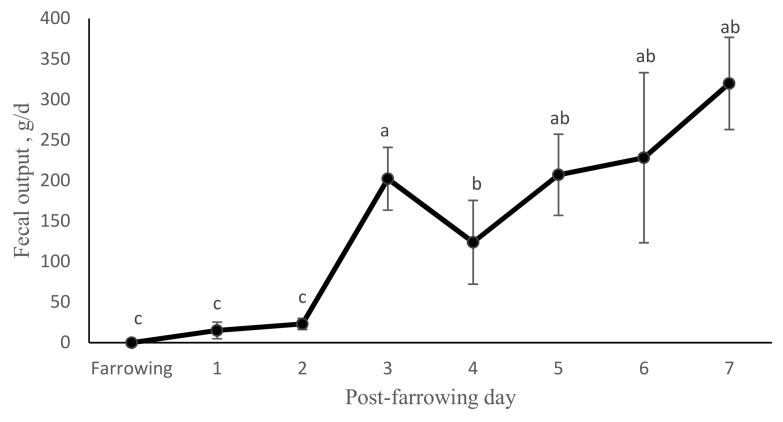
Fecal output (DM basis) ± standard errors (SE) of sows (*n* = 8) after farrowing. The data presented are least square means ± SE and represent the best estimate of means that was obtained using repeated measurements. Days with different superscripts are significantly different from each other based on Friedman test (*p* < 0.05).

**Table 1 animals-09-00268-t001:** Nutrient content and composition of sow diet.

**Diet Composition (as Fed)**	
Corn, %	56.80
SBM, %	30.00
Vitamin pre-mix, %	3.00
Molasse, %	3.65
Tallow, %	2.00
Salt, %	0.35
Calcium carbonate, %	0.70
Dicalcium phosphate, %	2.50
Potassium chloride, %	0.50
Pellet binder, %	0.50
**Analyzed Composition ^*^ (DM basis)**	
DM, %	83.40
CP, %	22.00
ADF, %	4.50
NDF, %	8.00
Fat, %	5.06
Ashe, %	7.59
Ca, %	1.29
P, %	1.07
Mg, %	0.24
K, %	1.44
Na, %	0.17
Fe, ppm	650
Zn, ppm	189
Cu, ppm	24.0
Mn, ppm	144
Mo, ppm	4.00

DM: dry matter; CP: crude protein; ADF: Acid detergent fiber; NDF: Neutral detergent fiber; ^*^ Feed sample analyzed by Dairy One, Inc. laboratory.

**Table 2 animals-09-00268-t002:** Behavior definitions.

Behavior	Definition
Laying	Lying on one side, four legs, or sternum
Standing	Straight up on four legs without moving any leg
Active	Walking or running through the pen. Sniffing, touching, or rubbing other piglets or pen walls or floor
Fighting	Biting or pushing another piglet
Nursing	From the moment 90% of the litter stop massaging the udder, remain still and began suckling until the time 10% of the litter start massaging the udder again or stop suckling.
Interacting with maternal feces	The amount of time piglets within a litter spent rooting, sniffing, touching, eating, or playing with maternal feces

**Table 3 animals-09-00268-t003:** Analyzed nutrient contents (DM basis) of lactating sow feces.

Analyzed Composition ^*^	(*n* = 8)	SE
DM, %	37.5	0.55
CP, %	21.1	0.43
TP, %	16.3	0.98
ADF, %	7.95	0.62
NDF, %	21.4	1.41
Fat, %	3.99	0.30
Ashes, %	31.7	0.60
Ca, %	6.36	0.17
P, %	4.76	0.15
Mg, %	1.46	0.04
K, %	0.64	0.06
Na, %	0.14	0.02
Fe, ppm	4039	137
Zn, ppm	1257	40.2
Cu, ppm	244	11.8
Mn, ppm	990	29.2
Mo, ppm	5.17	0.26

^*^ Mean value of each nutrient; SE: standard error of the means; DM: dry matter; CP: crude protein; TP: True protein; ADF: Acid detergent fiber; NDF: Neutral detergent fiber.

**Table 4 animals-09-00268-t004:** Control (CON) and deprived (DEP) litter behaviors at 7 d of age.

Behavior	Treatments		SE	*p*-Value
	CON (*n* = 3)	DEP (*n* = 4)		
Nursing duration, s	66.7	62.3	4.46	0.49
Nursing interval, m	42.6	42.4	3.11	0.95
Nursing per hour	1.38	1.40	0.12	0.89
Laying, %	73.1	72.2	2.00	0.70
Active, %	7.00	5.00	1.00	0.22
Standing, %	5.60	5.20	1.20	0.82
Fighting, m/d	5.41	6.08	3.42	0.89
Feces interaction, m/d	4.27	-	0.51	-

**Table 5 animals-09-00268-t005:** Survival rate and complete blood cell count in piglets with (CON) or without (DEP) access to maternal feces at birth, 7, and 21 d of age.

Dependent Variables	Birth (*n* = 8)		7 d (*n* = 8)		21 d (*n* = 8)		SE ^1^	*p*-Value ^2^			LS ^3^
	CON	DEP	CON	DEP	CON	DEP		TRT	Time	Interaction	
Litter Size	15.2 ^a^	15.7 ^a^	13.2 ^b^	12.5 ^b^	13.0 ^b^	11.7 ^b^	1.25	0.77	<0.01	0.46	-
Mortality rate, %	-	-	16.0	19.0	16.1	19.1	0.83	0.77	0.61	0.73	<0.01
Survival rate, %	-	-	84.0	81.2	83.9	80.9	7.54	0.80	0.54	0.72	<0.01
WBC, 10^9^/L	16.2 ^d^	9.79 ^e^	10.3 ^d^	10.1 ^e^	12.6 ^d^	9.52 ^e^	1.25	<0.01	0.18	0.12	0.05
Lymphocyte, 10^9^/L	2.89 ^b^	3.03 ^b^	6.32 ^a^	5.79 ^a^	8.09 ^a^	6.06 ^a^	0.70	0.217	<0.01	0.26	0.48
Monocyte, 10^9^/L	0.115 ^b^	0.130 ^b^	0.766 ^a^	0.775 ^a^	0.212 ^b^	0.456 ^b^	0.14	0.44	<0.01	0.63	0.66
Neutrophil, 10^9^/L	13.16 ^u^	6.60 ^v^	3.21 ^w^	3.54 ^vw^	4.34 ^vw^	3.07 ^w^	1.07	<0.01	<0.01	0.01	0.02
NLR	4.69 ^u^	2.15 ^v^	0.534 ^w^	0.637 ^w^	0.543 ^w^	0.490 ^w^	0.21	<0.01	<0.01	<0.01	0.05
RBC, 10^9^/L	6.24 ^b^	5.51 ^b^	4.95 ^c^	5.05 ^c^	6.99 ^a^	6.98 ^a^	0.37	0.61	<0.01	0.16	0.22
Hemoglobin, g/dL	10.70 ^b^	10.42 ^b^	7.42 ^c^	8.48 ^c^	12.54 ^a^	12.66 ^a^	0.56	0.63	<0.01	0.14	0.25
Hematocrit, %	35.48 ^b^	34.40 ^b^	22.93 ^c^	26.22 ^c^	40.15 ^a^	40.55 ^a^	1.95	0.67	<0.01	0.22	0.21
Mean corpuscular volume, fl	57.42 ^a^	62.45 ^a^	45.94 ^b^	51.68 ^b^	57.52 ^a^	57.92 ^a^	2.00	0.07	<0.01	0.27	0.65
Mean corpuscular hemoglobin, pg	17.3 ^a^	19.0 ^a^	15.0 ^b^	16.8 ^b^	18.0 ^a^	18.1 ^a^	0.69	0.138	<0.01	0.18	0.53

^a,b,c^ Means with different subscripts differ due to time effect (*p*-values < 0.05); ^d,e^ Means with different subscripts differ due to treatment effect within time (*p*-values < 0.05); ^u,v,w^ Means with different subscripts differ due to time × treatment interaction (*p*-values < 0.05); ^1^ Largest SE of least squares means; ^2^ Significant level for main effect of treatment, time, and treatment × time interaction; ^3^ Significant level of LS as a covariate.

**Table 6 animals-09-00268-t006:** Pre and post-weaning performance of piglets with (CON) and without (DEP) access to maternal feces.

Item	CON	DEP	SE	*p*-Value ^1^
Body Weight, kg	(*n* = 4)	(*n* = 4)		
Birth	1.45	1.30	0.150	0.94
7 d of age	2.72	2.37	0.141	0.845
21 d of age	6.45	6.07	0.146	0.83
Weaning	7.75	7.43	0.140	0.85
7 d post-weaning	9.21	8.54	0.175	0.71
14 d post-weaning	11.8	10.9	0.287	0.65
21 d post-weaning	15.53	13.98	0.442	0.40
28 d post-weaning	18.7	17.2	0.543	0.42
56 d post-weaning	33.5	29.9	1.23	0.058
73 d post-weaning	48.8 ^a^	43.4 ^b^	1.65	<0.01
102 d post-weaning	75.0 ^a^	66.7 ^b^	1.89	<0.01
123 d post-weaning	100.7 ^a^	91.4 ^b^	2.29	<0.01
ADG, kg/d				
7 d of age	0.18	0.15	0.014	0.49
21 d of age	0.27	0.26	0.014	0.86
Weaning	0.30	0.30	0.014	0.99
7 d post-weaning	0.21	0.16	0.019	0.25
14 d post-weaning	0.37	0.35	0.022	0.55
21 d post-weaning	0.54 ^a^	0.43 ^b^	0.027	0.020
28 d post-weaning	0.45	0.45	0.028	0.87
56 d post-weaning	0.52	0.44	0.028	0.10
73 d post-weaning	0.89	0.81	0.033	0.12
102 d post-weaning	0.90 ^a^	0.76 ^b^	0.041	<0.01
123 d post-weaning	1.23	1.16	0.062	0.15
ADFI, kg/d				
7 d post-weaning	0.27	0.23	0.016	0.15
14 d post-weaning	0.45	0.43	0.017	0.73
21 d post-weaning	0.76	0.70	0.045	0.32
28 d post-weaning	0.84	0.78	0.047	0.34
56 d post-weaning	1.41	1.29	0.038	0.058
73 d post-weaning	2.25 ^a^	2.01 ^b^	0.059	<0.01
F:G				
7 d post-weaning	1.33	1.48	0.091	0.31
14 d post-weaning	1.22	1.25	0.041	0.86
21 d post-weaning	1.41	1.64	0.060	0.13
28 d post-weaning	1.87	1.76	0.141	0.44
56 d post-weaning	2.69	2.87	0.119	0.20
73 d post-weaning	2.52	2.46	0.099	0.62
Overall performance				
^*^ADG (kg/day)	0.54 ^a^	0.48 ^b^	0.008	<0.01
^+^ADFI (kg/day)	0.99 ^a^	0.90 ^b^	0.023	0.02
^+^F:G	1.84	1.91	0.071	0.36

SE = Largest SE of least squares means; ^a,b^ Within a row, least squares means without a common superscript differ (*p* < 0.05) due to treatment effect; ^*^ Overall ADG from birth until 123 d post-weaning; ^+^ Overall ADFI and F:G ratio from weaning until 73 d post-weaning.

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
