# Peer review of "Preliminary Study: Depriving Piglets of Maternal Feces for the First Seven Days Post-Partum Changes Piglet Physiology and Performance before and after Weaning"

_animals, 2019, doi:10.3390/ani9050268_

Round 1
Reviewer 1 Report
Abstract: Add the actual P-value instead of greater of less than 0.05.
Introduction:
Line 44: What kind of nutrients are there is feces?
line 48: name some of those essential bacteria?
Results:
Line 165: Is it the actual P-value? If not add it.
Line 169: Do the same as line 165
Table 3: are these numbers the average numbers? It needs clarification.
Table 4: The last column is messed. Please revise it.
sections titled: "piglet hematology and survival" and "pre and post weaning growth performance": rewrite the P-values and add actual P-values.
Discussion:
Line 257: How much feces was consumed by piglets in this study?
Author Response
Response to Reviewer 1 Comments
Point 1bstract: Add the actual P-value instead of greater of less than 0.05.
P values were changed to the actual value.
Introduction:
Point 2 Line 44: What kind of nutrients are there is feces?
Feces contain unabsorbed nutrients from the feed and from endogenous sources. As nutritional analysis showed (table 3), lactating sow feces contain nutrients like protein (undigested, microbial, and endogenous) fiber, volatile and non-volatiles fatty acids, and minerals ( Ca, P, Zn, Cu Fe etc. ) that could be beneficial to piglets. We added this in the manuscript for clarification (line 48-49)
Point 3 line 48: name some of those essential bacteria?
Even when there are several studies that evaluate foal microbiome, to our knowledge, a specific positive bacterium that is transfer from mare to foal via coprophagy haven’t been identified. The two papers cited agree that coprophagy is a method to acquire beneficial bacteria, but they did not mention which bacteria are in feces.
Results:
Point 4 Line 165: Is it the actual P-value? If not add it.
The actual p value is < 0.001. Since we are reporting the data using only two decimals we wrote it as < 0.01.
Point 5 Line 169: Do the same as line 165
The actual p value is <0.001, It was corrected in the document
Point 6 Table 3: are these numbers the average numbers? It needs clarification.
Yes, the numbers are the mean of each nutrient. This was added in the footnotes for clarification.
Point 7 Table 4: The last column is messed. Please revise it.
Dear reviewer we don’t understand what you mean. The las column is reporting the P value for each behavior. Please be more specific so we can address your concern.
Point 8 sections titled: "piglet hematology and survival" and "pre and post weaning growth performance": rewrite the P-values and add actual P-values.
The exact p values were reported as requested. In sentences where we are talking about more than one variable, p < 0.05 is written to prevent readers from confusion. Notwithstanding, all the exact p values are reported in the tables.
Discussion:
Point 9 Line 257: How much feces was consumed by piglets in this study?
In this study we did not measure the amount of feces consumed by piglets. However, in this sentence we are talking about sows feed intake. We are discussing a possible explanation for the low fecal output sows had the first 2 days post-partum.
The objective of the study was to evaluate the effects of fecal deprivation and not to determine fecal consumption. As mentioned in the manuscript, piglet fecal consumption was estimated by Samsom and Gleed (1981). The main objective of the study was to evaluate the effect of fecal deprivation by piglets and not to determine how much feces piglets consume.
Reviewer 2 Report
General Comments
The concept behind this paper is very intriguing and will be of interest to fields of nutrition and animal production. However, I feel calling it a Pilot Study is slightly misleading and should maybe be changed to Preliminary Study, as in the conclusion. There are consistent minor mistakes in language throughout. I had begun to correct these in the detailed comments but it needs to be proof-read by a native English speaker if possible. I would like to see this paper published, but feel certain aspects of the manuscript need developing. I hope that my comments are useful and can help to improve the reporting of the study. I look forward to reading the revised manuscript again.
Introduction
The introduction sets the question nicely but is fairly brief. The title makes mention of the fact this is a pilot study and I feel like this needs to be addressed in the introduction too. I.e. a justification for why the results are being published at this stage and not when more data have been collected?
Methods
In places these are really too vague and need fleshing out. There is no mention of specifics of behavioural analysis such as how many pigs were observed, whether the pigs were filmed or scored live and if any observation software was used.
Results
There are very few test statistics reported. I have added more detailed comments about the results below.
Detailed Comments
Line 54: Typo “from eating”
57: Typo “Based”
59: Grammar: “compared to the ones”
87: Please explain the rationale behind the 7 day time point deprivation?
Table 1:
· This table contains important information but is a little confusing for the reader. It’s set out in a fairly unconventional way and I would suggest reformatting so that the diet composition is on top and the nutrient content underneath. For format example, please see a paper such as: Laird et al., 2018. Super-dosing phytase improves the growth performance of weaner pigs fed a low iron diet, Animal Feed Science and Technology, Volume 242, Pages 150-160.
· It also needs to be specified whether the nutrient content is “as-fed” or is it % of the dry matter?
· Please specify what the fat in the diet was
· “Ash” not “ashes”
· Finally, there are some typos in the footnotes of the table. Please correct to “Crude protein”, “Neutral detergent fiber”. TP is also included in the notes but is not in the table?
93-95: Please expand on how the faeces were collected without depriving the piglets in the control group? Was this done at the end of the day when the piglets were sleeping?
99-100: Please clarify if this is the normal time at which these procedures would be carried out or whether this was altered for the study.
Behaviour: Was this filmed or scored in person? If it was filmed, please provide details of the filming set-up. How many piglets were observed for the lying, standing, active and fighting behaviours? How was pushing in the fighting category distinguished from piglet play behaviour?
141-142: I cannot see details of the grower-finisher unit above?
142-143: Are the monthly measures of growth performance the ADFI, ADG and F:G? If so, please state this for clarity.
153-154: It would be useful to see the difference in AIC values for the models if possible.
154-55: I do not understand the sentence about the Fisher’s LSD, which period is being referred to?
159-160: Given the recent shift away from reliance on p-values (see: https://www.nature.com/articles/d41586-019-00857-9), I would tend to caution against using p-value ‘tendency’ values.
Table 3: Correct typos
199: 4 minutes of interaction with faeces seems very low. How did you determines if the piglets consumed the faeces? There was no before and after weight and consumption is not reported as a stand-alone behaviour.
Table 4: Why are some behaviours in m/d and some in %? Can you please clarify what the percentage is of – is it of 24h? Were the percentage data analysed with the one-way ANOVA?
Line 216: close bracket
219-22: HCT values – could these have been explained by differences in piglet hydration status? Were bloods collected at the same time of day/similar time after feeding to account for this?
Table 5: These results are slightly confusing. Many of the variables seem unsuitable for an ANOVA analysis (e.g. percentage data, data bounded by zeros). In addition, I do not understand how litter size can have an effect on survival but not mortality, when surely these variables are inherently linked. In addition, I would suggest changing the heading Parameter to Dependent Variable, as a parameter is estimated and a dependent variable is measured.
Table 6: These production results are very interesting but there are almost 40 individual tests in this table. These really need to be corrected for multiple comparisons or analysed in a different way as the likelihood of finding a significant result is hugely increased using the current method. Conducting this many individual ANOVAs to test the same hypothesis is unadvisable. Could you maybe add time to the model as in the previous repeated measures analysis?
252-3: Can you include this number in the results section?
257: The litter size stated here is considerably smaller than the litters reported in the paper. Why is a LS of 10 being assumed?
272-3: When were treatment groups bled?
278-9: Please provide references for this.
288-94: Are these studies that you’re currently working on or are you referring to the results of the current study? It seems odd to include this here in passing.
Conclusions
334-5: I think the authors need to be careful here and I would remove this sentence. Statistically meaningful is not directly related to the results being significant. Please see articles such as https://journals.sagepub.com/doi/full/10.1177/0956797611417632 and https://journals.sagepub.com/doi/full/10.1177/1745691614551642.
Author Response
Response to Reviewer 2 Comments
The concept behind this paper is very intriguing and will be of interest to fields of nutrition and animal production. However, I feel calling it a Pilot Study is slightly misleading and should maybe be changed to Preliminary Study, as in the conclusion. There are consistent minor mistakes in language throughout. I had begun to correct these in the detailed comments but it needs to be proof-read by a native English speaker if possible. I would like to see this paper published, but feel certain aspects of the manuscript need developing. I hope that my comments are useful and can help to improve the reporting of the study. I look forward to reading the revised manuscript again.
Dear reviewer thanks for all your comments. They have been very helpful to improve this manuscript. The manuscript was revised by a native English speaker editor. Please find below our responses to your comments.
Introduction
Point 1The introduction sets the question nicely but is fairly brief. The title makes mention of the fact this is a pilot study and I feel like this needs to be addressed in the introduction too. I.e. a justification for why the results are being published at this stage and not when more data have been collected?
We agree that the best way to describe our study is as a preliminary study. A pilot study may assume a continuation of it. This was a small study conducted as part of an ongoing research in swine semiochemicals. Currently we don’t have the resources or the time to collect more data. However, the findings open the door for multiple research in the field of swine nutrition, production, semiochemicals and microbiome. Although brief, we believe this is a significant research and its results should be disseminated for the benefit of researchers and the swine industry.
Methods
Point 2 In places these are really too vague and need fleshing out. There is no mention of specifics of behavioural analysis such as how many pigs were observed, whether the pigs were filmed or scored live and if any observation software was used.
To evaluate piglet’s behavior video cameras were placed on the top of each farrowing crate (picture below). The entire litter was observed during the 24 h period. No software was used to code the data. A single observer watched the videos using Windows media player and recorded the data in excel. More details were added to the manuscript in regard to behavioral observations. Please see line 142-149 in the new document.
Results
Point 3 There are very few test statistics reported. I have added more detailed comments about the results below.
Test statistic were added to the results section.
Detailed Comments
Point 4 Line 54: Typo “from eating”
Corrected in the manuscript
Point 5 57: Typo “Based”
Corrected in the manuscript
Point 6 59: Grammar: “compared to the ones”
Corrected in the manuscript
Point 7 87: Please explain the rationale behind the 7 day time point deprivation?
If no extraneous iron is provided, piglets will start showing signs of anemia at 7-14 d of age because of their rapid growth rate, rapid increase in blood volume, and the low iron content of sow milk (Furugouri, 1975; Svoboda and Drábek, 2005; Victor and Mary, 2012). Based on this, if coprophagy have an effect on piglet’s hemoglobin, RBC, hematocrit etc. we were expecting to see hematological differences at this time point. This also explain why we did not castrate or injected piglets with iron until 7 d of age. We processed the litters and stop fecal deprivation at 7 d and since further fecal or iron deprivation could have compromised piglet’s welfare and health. In addition, studies have shown that piglets’ preference towards maternal feces is not significant after 7 d (Horrel and Hodgson,1992). Because of this, we were expecting to see some difference by seven days of age. This explanation was added to the discussion (lines 282-285) of the new document.
Point 8 Table 1:
· This table contains important information but is a little confusing for the reader. It’s set out in a fairly unconventional way and I would suggest reformatting so that the diet composition is on top and the nutrient content underneath. For format example, please see a paper such as: Laird et al., 2018. Super-dosing phytase improves the growth performance of weaner pigs fed a low iron diet, Animal Feed Science and Technology, Volume 242, Pages 150-160.
We formatted the table as suggested
Point 9 · It also needs to be specified whether the nutrient content is “as-fed” or is it % of the dry matter?
The nutrient content is as % of DM but diet composition is as fed. We added in the title of the table
Point 10· Please specify what the fat in the diet was
The fat used in the diet was Tallow
Point 11· “Ash” not “ashes”
Corrected
Point 12· Finally, there are some typos in the footnotes of the table. Please correct to “Crude protein”, “Neutral detergent fiber”. TP is also included in the notes but is not in the table?
Typos are corrected. TP is part of the fecal analysis but not of the feed. It was deleted.
Point 1393-95: Please expand on how the faeces were collected without depriving the piglets in the control group? Was this done at the end of the day when the piglets were sleeping?
Only 5g of feces were collected daily. Thus, piglets in the control group had access to more than 98 % of the feces sow excreted (based on the observed fecal output). All fecal samples were collected around 6:00 am, when sows were fed, since they tend to defecate while eating or shortly after eating. After a control group sow defecated, the observer weighed the feces, collected the five grams and placed them back in the crate. This was done only once a day. If sow defecated more than once, all feces were placed on the pen after weighed. At 6:00 am most of the litter were sleeping, however they tended to wake up when sow stand up to eat. Clarifications can be found in line 105 of the new document.
Point 14 99-100: Please clarify if this is the normal time at which these procedures would be carried out or whether this was altered for the study.
This is not the normal time these procedures are conducted. They are usually conducted when piglets are 3-5 days old. However, in order to evaluate the effects of fecal deprivation on piglet’s hematology, we did it at 7 days of age since, if no extraneous iron have been given, at this time signs of anemia can be detected. Delaying these procedures was necessary to evaluate the effects of fecal deprivation on piglets hematology.
Point 15Behaviour: Was this filmed or scored in person? If it was filmed, please provide details of the filming set-up. How many piglets were observed for the lying, standing, active and fighting behaviours? How was pushing in the fighting category distinguished from piglet play behaviour?
As mentioned above, litters behavior was filmed and all piglets within a litter were observed. We did not evaluate playing behavior in this study. At this age, most of the aggression occurs when piglets are nursing. The observed pushing incidents recorded as aggression were always accompanied with biting which makes it easy to distinguish it from playing. Playing was included as an active behavior.
Point 16 141-142: I cannot see details of the grower-finisher unit above?
Raised pens measuring 1.5 x 3 m, with slat plastic floor were used at weaning. Finishing pens were 3.65 x 2.15 m with slat concrete floor. Details were added to the manuscript. Please see lines 155-156
Point 17 142-143: Are the monthly measures of growth performance the ADFI, ADG and F:G? If so, please state this for clarity.
It was added to the manuscript. Please see lines 162-163
Point 18 153-154: It would be useful to see the difference in AIC values for the models if possible.
The biggest difference in AIC value was for piglet weight. The AIC value when the model included a first order autoregressive co-variance structure was 292. If no covariance structure was added to the model the AIC value was 349. For the other variables a first order autoregressive co-variance structure produced the lowest AIC but the differences were not as significant as this.
Point 18 154-55: I do not understand the sentence about the Fisher’s LSD, which period is being referred to?
As mentioned in the statistic section of the manuscript, data collected over time were analyzed as a repeated measure. The model included the effect of time, treatment and their interaction. The repeated measure ANOVA will give us a significant value for the main effect of treatment, time, and their interaction. Fisher’s LSD was used to evaluate any treatment effect within time (i.e. if there was a treatment effect at 7 d or 21 d post-weaning). We did this so the reader could see if at a particular time point the measurements were different. However, our main conclusions are based on the main effect of treatment (overall performance).
Point 19 159-160: Given the recent shift away from reliance on p-values (see: https://www.nature.com/articles/d41586-019-00857-9), I would tend to caution against using p-value ‘tendency’ values.
Thank you for the reference, it was very informative. The sentence was deleted.
Point 20 Table 3: Correct typos
Corrected
Point 21 199: 4 minutes of interaction with faeces seems very low. How did you determines if the piglets consumed the faeces? There was no before and after weight and consumption is not reported as a stand-alone behaviour.
In this study we did not quantified fecal consumption. As mentioned in the introduction and discussion, fecal consumption was estimated in a previous study (Samson and Gleed, 1981). Since this and other studies in the literature provide evidence that piglets consume maternal feces, our goals were to evaluate the effects of fecal deprivation on piglet’s hematology and performance. We measured this behavior since we wanted to know how much time they are actually interacting with feces and because this behavior has not been reported in the literature. Thus, this is the first study that actually observed and quantify this behavior.
Point 22 Table 4: Why are some behaviours in m/d and some in %? Can you please clarify what the percentage is of – is it of 24h? Were the percentage data analysed with the one-way ANOVA?
Low frequency behaviors such as fighting, nursing, and interacting with feces were coded using a continuous sampling technique while high frequency behaviors were measured using a scan sampling technique.
A continues sampling consist of measuring the amount of time a behavior was observed during the 24 h period . Thus, the units are m/d. On the other hand, on a scan sampling technique the observer evaluates litters behavior at predetermine time intervals (every 10 minutes in this study). Thus, when this technique is used behavior measurements are expressed as the percentage of time piglets were observed doing a particular behavior across the number of observation (144 observations per litter in the 24 h period).
Both techniques are valid ways to study pig behavior (Mitlohner et al., 2001). The difference in unit is due to the different techniques used.
Yes, the percentage data were analyzed using ANOVA. Usually behavior data needs to be transformed to meet parametric assumption, but in this study no transformation was required since raw data was parametric (normal distribution and homogeneity of variance). This is an acceptable way to analyze behavior data.
Point 23 Line 216: close bracket
Corrected in the document
Point 24 219-22: HCT values – could these have been explained by differences in piglet hydration status? Were bloods collected at the same time of day/similar time after feeding to account for this?
Blood samples were collected at the same time of day (around 7:00 am).
Point 25 Table 5: These results are slightly confusing. Many of the variables seem unsuitable for an ANOVA analysis (e.g. percentage data, data bounded by zeros). In addition, I do not understand how litter size can have an effect on survival but not mortality, when surely these variables are inherently linked. In addition, I would suggest changing the heading Parameter to Dependent Variable, as a parameter is estimated and a dependent variable is measured.
The heading was changed as suggested. Mortality was not affected by LS because the data was showing the number of animals that died not the mortality rate. To keep the data consistent, we changed mortality ( number of animals) to mortality rate (%). After doing this the covariate was significant. The table was modified in the manuscript. Thank you for this observation.
Since hematological, survival rate and mortality rate data were normally distributed and had homogeneity of variance (based on Levene test) an ANOVA is a valid test since data meet parametric assumptions. Several studies have used ANOVA to assess hematological differences (Brown et al. 1996; Chewen et al. 2001, Rolinec et al. 2015).
Point 26 Table 6: These production results are very interesting but there are almost 40 individual tests in this table. These really need to be corrected for multiple comparisons or analyzed in a different way as the likelihood of finding a significant result is hugely increased using the current method. Conducting this many individual ANOVAs to test the same hypothesis is unadvisable. Could you maybe add time to the model as in the previous repeated measures analysis?
As mentioned in the manuscript and in an above comment, data were analyzed as a repeated measure. The effect of time was including in the model as well as the time treatment interaction. The table is showing the comparison within period using the obtained LSD from the ANOVA output of SAS. The main effect of treatment is shown in the overall.
Although by the way the table is written you could assume that we did multiples ANOVA, the data was not analyzed multiple times. From the repeated measures ANOVA output we obtained the within time comparison. The “pdiff” command in the SAS code gave us the multiple comparisons of treatment within time. Thus, everything was obtained from one analysis.
The purpose of this table is to show the readers the performance across the entire study. A similar analysis and table were used by McGlone et al. (2017). If you think the table is confusing and that the data should be presented in a different way, please let us know and we will change it as requested.
Point 27 252-3: Can you include this number in the results section?
It was added (line 192)
Point 28 257: The litter size stated here is considerably smaller than the litters reported in the paper. Why is a LS of 10 being assumed?
According to the latest pork-check of statistic, the number of piglet per litter in 2017 was 10.60 (https://www.pork.org/facts/stats/structure-and-productivity/productivity-measures-of-u-s-pig-herds/). This is in agreement with the 2015 USDA Overview of the United states hog industry report.
Point 29 272-3: When were treatment groups bled?
Since three out of the four sows in this group farrowed at night (10:00 pm – 6: 00 am) blood samples were collected around 7:00 am the next day. During nighttime, only one observer was at the farm and at least two people are needed to bleed piglets. Thus, piglets within these litters were bled when another person was available to assist in blood collection.
Point 30 278-9: Please provide references for this.
Reference were added
Point 31 288-94: Are these studies that you’re currently working on or are you referring to the results of the current study? It seems odd to include this here in passing.
We are speculating about why maternal feces could be beneficial to piglets. We are currently working whit these possible hypotheses. So far, we were able to identify possible maternal semiochemicals in lactating sow feces. The use of these semiochemicals seams to improve performance and reduced aggression at weaning (unpublished data; studies are in process). What we are trying to do is discuss some possible mechanisms through which piglet performance can be improved by having access to maternal feces. Our hypothesis needs to be further evaluated. However, we want to give the readers ideas for possible future studies. The next paragraphs support our speculations.
Conclusions
Point 32 334-5: I think the authors need to be careful here and I would remove this sentence. Statistically meaningful is not directly related to the results being significant. Please see articles such as https://journals.sagepub.com/doi/full/10.1177/0956797611417632 and https://journals.sagepub.com/doi/full/10.1177/1745691614551642.
Dear reviewer, thanks for these two-reference papers. After reading the paper we agree with you and we deleted the sentence .

Round 2
Reviewer 2 Report
I enjoyed reading the revised manuscript and thank the authors for the time they have taken to make the changes. I would be happy to see this paper published and think it makes a valuable contribution to the literature.
My only comments are related to the stats. Although I now understand Table 6 more clearly, the multiple LSD tests still have the issue of increasing the chance of a false positive. The default pdiff command in SAS does not adjust for multiple comparisons. I would suggest either correcting for multiple comparisons or stating in the text or table that the pairwise tests were not corrected. For Table 5, I think you only need either survival rate or mortality rate and not both (because they are inversely linked). Additionally, there is a slight issue with behavioural data measured as percentages and analysed by ANOVA. Although this is widely done, it isn’t necessarily the best suited option. Count data models could be used for rates of behaviour, or binomial models for proportions. Although these are my suggestions, it is the decision of the author on how they would like to present the data.